

# Curcumin plays a synergistic role in combination with HSV-TK/GCV in inhibiting growth of murine B16 melanoma cells and melanoma xenografts

Hong Li[1,2,*], Haiyan Du[1,2,*], Guangxian Zhang[1], Yingya Wu[1], Pengxiang Qiu[1], Jingjing Liu[3], Jing Guo[3], Xijuan Liu[1], Lingling Sun[4], Biaoyan Du[2,3] and Yuhui Tan[1,2]

[1] Department of Biochemistry and Molecular Biology, Guangzhou University of Chinese Medicine, Guangzhou, China
[2] The Research Center of Basic Integrative Medicine, Guangzhou University of Chinese Medicine, Guangzhou, China
[3] Department of Pathology, Guangzhou University of Chinese Medicine, Guangzhou, China
[4] Integrative Cancer Center, First Affiliated Hospital, Guangzhou University of Chinese Medicine, Guangzhou, China
* These authors contributed equally to this work.

Corresponding authors
Biaoyan Du,
dubiaoyan@gzucm.edu.cn
Yuhui Tan, yhuitan@foxmail.com

## ABSTRACT

Melanoma is a global concern and accounts for the major mortality of skin cancers. Herpes simplex virus thymidine kinase gene with ganciclovir (HSV-TK/GCV) is a promising gene therapy for melanoma. Despite its low efficiency, it is well known for its bystander effect which is mainly mediated by gap junction. In this study, we found that curcumin reduced B16 melanoma cell viability in both time- and dose-dependent manner. Further study showed that curcumin improved the gap junction intercellular communication (GJIC) function, and upregulated the proteins essential to gap junction, such as connexin 32 and connexin 43, indicating the potential role in enhancing the bystander effect of HSV-TK/GCV. By co-culturing the B16$^{TK}$ cells, which stably expressed TK gene, with wildtype B16 (B16$^{WT}$) cells, we found that co-treatment of curcumin and GCV synergistically inhibited B16 cell proliferation, but the effect could be eliminated by the gap junction inhibitor AGA. Moreover, curcumin markedly increased apoptosis rate of B16$^{WT}$ cells, suggesting its effect in enhancing the bystander effect of HSV-TK/GCV. In the in-vivo study, we established the xenografted melanoma model in 14 days by injecting mixture of B16$^{TK}$ and B16$^{WT}$ cell in a ratio of 3:7. The result demonstrated that, co-administration of curcumin and GCV significantly inhibited the xenograft growth, as indicated by the smaller size and less weight. The combinational effect was further confirmed as a synergistic effect. In conclusion, the results demonstrated that curcumin could enhance the killing effect and the bystander effect of HSV-TK/GCV in treating melanoma, which might be mediated by improved gap junction. Our data suggested that combination of HSV-TK/GCV with curcumin could be a potential chemosensitization strategy for cancer treatment.

# INTRODUCTION

Melanoma is one of the cancers for which the incidence is still on the rise worldwide (*Bray et al., 2018*). Although melanoma only accounts for less than 5% of skin cancers, it leads to the majority of deaths from skin cancers (*Ferlay et al., 2019*). Much attention has been attracted to improve the therapeutic strategy of this cancer (*Prado, Svoboda & Rigel, 2019*; *Wrobel, Przybylo & Stepien, 2019*). To date, for melanoma at early stage, surgical resection is a critical treatment. However, in most cases, it progresses into advanced stage and becomes resistant to conventional treatments (*Heo et al., 2016*). Hence, it is urgent to develop new therapies to counteract the limitation of conventional therapy and to slow down the pathological process.

During the last two decades, gene therapy has emerged as a promising alternative to treat a variety of human malignancies (*Kumar et al., 2016*). Suicide gene therapy is such a sort of gene therapy introducing the heterogeneous suicide gene that is able to converse a non-toxic prodrug into a lethal drug. Among various suicide systems, herpes simplex virus thymidine kinase gene with ganciclovir (HSV-TK/GCV) is one of the most in-depth-studied systems and has been introduced to treat melanoma (*Navarro et al., 2016*). In this system, viral thymidine kinase is expressed and subsequently metabolizes the prodrug GCV to mono-phosphorylated GCV, which will be further converted into triphosphate. Since GCV triphosphate is an analogue of deoxyguanosine triphostphate, it consequently inhibits DNA synthesis, causing the tumor cell death (*Duarte et al., 2012*). Of note, increasing evidence showed that the bystander effect, which is known to induce tumor regression despite that only a percentage of cancer cells express the TK gene, is a critical mechanism contributing to the anti-tumor effect of HSV-TK/GCV therapy (*Van Dillen et al., 2002*). Since the efficiency still remains a challenge to HSV-TK/GCV therapy and it limits the clinical application, numerous studies have been focused on improving its bystander effect (*Li et al., 2011*; *Rautsi et al., 2008*; *Xiao et al., 2018*).

Curcumin is a bioactive component of *Curcuma Longa L*, which is a popular traditional medicine and culinary material in some countries (*Ammon & Wahl, 1991*). Due to its multiple biological properties, curcumin has been widely studied for its protective role in treating cancer, cardiovascular diseases, chronic inflammatory diseases, neurodegenerative diseases and arthritic (*Aggarwal & Harikumar, 2009*; *Li et al., 2019*; *Willenbacher et al., 2019*). Previous studies showed that curcumin treatment in melanoma cell lines or mice with melanoma xenograft demonstrated a growth-inhibited effect, and several cellular and molecular mechanisms have been implicated (*Mirzaei et al., 2016*; *Nabavi et al., 2018*). However, whether curcumin has a synergistic effect on HSV-TK/GCV therapy in melanoma still remains unknown. Therefore, in the present study, we investigated the role of curcumin in B16 cells treated with HSV-TK/GCV and its impact on the bystander effect, and further determined the synergistic effect in xenografted melanoma.

## MATERIALS & METHODS

### Cell culture

The murine malignant melanoma cell line B16 (wildtype, WT) was obtained from Sun Yat-Sen University (Guangzhou, China). Cells were cultured in RPMI-1640 (Gibco/Invitrogen, Burlington, Ontario, USA) supplemented with 10% fetal bovine serum (FBS) and 100 U/mL penicillin and streptomycin at 37 °C in a humidified 5% $CO_2$ atmosphere. $B16^{TK}$ cells, which are B16 cells that stably express fused protein of HSV-TK and green fluorescent protein (GFP), and $B16^{WT}$ cells that express red fluorescent protein (RFP) were generated in our lab. For curcumin treatment, 20 mM curcumin was freshly prepared in DMSO and then diluted into the desired concentrations with the medium containing 10% FBS, followed by ultrasonication for 1 min. The dilutions were used to treat the cells immediately to avoid degradation.

### MTT assay

Cells were seeded in 96-well plates at a density of 4,000 cells per well and incubated overnight, followed by treated with various treatments. Then 10 μL of 5 mg/ml MTT reagent (Sigma, St. Louis, MO, USA) was added to each well to incubate at 37 °C for 4 h, and the formazan crystals were dissolved in 150 μL dimethyl sulphoxide (DMSO) in each well. Absorbance values were measured at wave length of 490 nm with a 96-well microplate reader (BioRad, Hercules, CA, USA). Each experiment was performed using 6 replicate wells for each treatment. The results were normalized to cells incubated in control medium, which were considered 100% viable.

### Double-fluorescence dye transfer assay

B16 cells were treated with various concentrations of curcumin (Sigma, St. Louis, MO, USA) in duplicate for 48 h. One population of cells in each group was stained with red fluorescein CMTMR and green fluorescein Calcein (Molecular Probes, Eugene, OR, USA) for 1 h at 37 °C. These cells were referred to "donor cells", while the other population of cells that was not subjected to fluorescein treatment was referred to "recipient cells". After washed with PBS, the donor cells and recipient cells were digested respectively and mixed at a ratio of 1:99. The mixed cells were seeded in dishes for 3 h, and then digested for assessment by flow cytometry (BD Biosciences, San Jose, CA, USA).

### Analysis of bystander effect

$B16^{TK}$ cells and $B16^{WT}$ cells were mixed in 1:4 and seeded in 6- or 96-well plates. 24 h later, the cells were subjected to the following treatment, respectively: DMSO (negative control), GCV (15 μM), curcumin (10 μM or 20 μM), curcumin (10 μM or 20 μM) with GCV (15 μM).The cell viability was detected by MTT assay. To determine the rate of apoptosis, the cells were digested and fixed in 70% ethanol at 4 °C, followed by stained with Annexin V for flow cytometry analysis. Of note, before harvesting, pictures of cell morphology in each group were taken by fluorescence microscope.

## Western blot analysis

B16 cells were cultured in 6-well plates for 48 h, followed by treatment with curcumin (5 $\mu$M, 10 $\mu$M, 20 $\mu$M) for another 24 h. Then, cells were homogenized in RIPA lysis buffer supplemented with protease inhibitor cocktail (Sigma, St. Louis, MO, USA) and 1 mM PMSF. The lysates were incubated on ice for 30 min with a vortexing every 5 min, and centrifuged at 12,000 g for 30 min. The supernatants were harvested, and then the protein concentrations were determined by BCA protein assay kit (Thermo Fisher Scientific, Rockford, IL, USA). An equal amount of the proteins (30 $\mu$g) was subjected to 10% SDS-PAGE following the immunoblot procedure as demonstrated previously (*Lu et al., 2016*). Membranes were blocked with 5% nonfat milk in TBS containing 0.1% Tween 20 for 90 min, followed by incubation overnight at 4 °C with specific antibodies to Cx32 (1:5,000; ABconel, Wuhan, China), Cx43 (1:5,000; ABconel, Wuhan, China) or GAPDH (1:10,000, ABconel, Wuhan, China). Blots were developed with enhanced chemiluminescence HRP substrate (Millipore, Bedford, MA, USA) and detected by a Tanon detection system (Shanghai, China). The intensities of the blots were quantified with the NIH image J software.

## Animals

SPF-class C57BL6/J mice (equivalent numbers of males and females, weighing 18–22 g) were supplied by the Experimental Animal Center of Sun Yat-Sen University (Guangzhou, China). All protocols were carried out in accordance with the Guide for the Care and Use of Laboratory Animals (NIH Publication No. 85–23, revised 1996), and were approved by the Institute Research Medical Ethics Committee of Guangzhou University of Chinese Medicine, China (20180408).

## Animal models of xenografted melanoma

B16$^{TK}$ cells and B16$^{WT}$ cells were mixed at the indicated ratios and diluted with serum-free culture medium. A total of $2 \times 10^5$ cells in a final volume of 100 $\mu$L was injected into the right flanks of each C57BL/6J mouse. The mice with tumor were randomized into four groups ($n = 16$ mice per group): a saline control group, a group treated with GCV (50 mg/kg/day), a group treated with curcumin (100 mg/kg/day), and a group treated with both GCV (50 mg/kg/day) and curcumin (100 mg/kg/day). For curcumin preparation, it was initially dissolved in DMSO and then diluted into 10 g/L with saline solution, followed by ultrasonication for 1 min and administered immediately. Saline and curcumin were administratered intraperitoneally once a day during the study period of 14 days. GCV was administratted daily from day 7 till the end. Then the mice were sacrificed and the solid tumors were isolated to measure the volume and mass.

## Statistical analysis

Data were presented as means ± standard deviation (SD). By using SPSS 13.0, statistical analysis was performed with unpaired Student's $t$-test between two groups or one-way analysis of variance (ANOVA) followed by LSD test among various groups. In all cases, difference was considered statistically significant at $P < 0.05$.
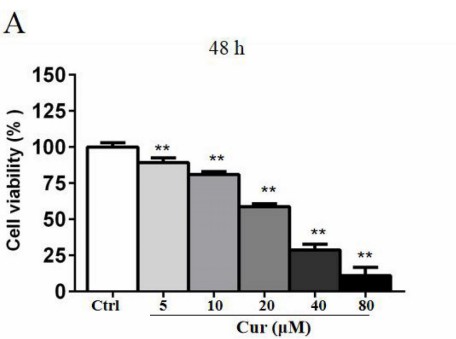
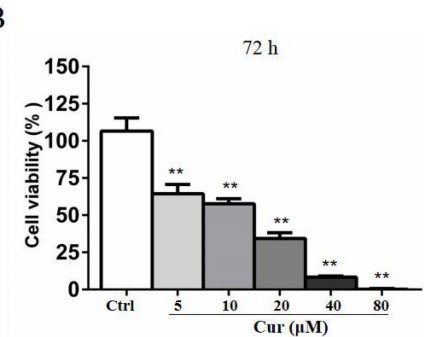

**Figure 1** **Curcumin has a cytotoxicity effect on B16 cells.** B16 cells were treated with or without curcumin (5 µM, 10 µM, 20 µM, 40 µM and 80 µM), and then the cell viability at 48 h (A) and 72 h (B) was determined by MTT assay, respectively. Data were presented as means ± SD. **$P < 0.01$ vs. ctrl group.

The combinational effect of two drugs was assessed by Q value using Zheng-Jun Jin's method (*Jin, 2004*). Briefly, $Q = EAB/[EA + EB(1 - EA)]$, in which EA, EB and EAB represent the effect of drug A, drug B and the combination of two drugs, respectively. According the Q value, the combinational effect can be deemed as an antagonistic effect ($Q < 0.85$), an additive effect ($0.85 < Q < 1.15$), or a synergistic effect ($Q > 1.15$).

# RESULTS

## Curcumin has a cytotoxicity effect on B16 cells

Curcumin is a natural compound that has been delineated to be toxic to various cancer cells, but the activity varies from cell types and depends on the time of incubation. To determine the the optimal concentration of curcumin in combination with HSV-TK/GCV treatment, we conducted MTT assay. B16 cells were treated with 5, 10, 20, 40, and 80 µM curcumin for 48 h or 72 h. As shown in Fig. 1, curcumin significantly inhibited B16 cells viability in both time- and dose-dependent manner. The half maximal inhibitory concentration (IC50) of curcumin in B16 cells was about 30 µM at 48 h and 17.3 µM at 72 h, respectively. Notably, a lower dose of curcumin (5 µM, 10 µM and 20 µM) for 48 h, which demonstrated relatively lower inhibitory effects on B16 cell viability, was used in other *in-vitro* studies to determine the effect of curcumin combined with HSV-TK/GCV treatment.

## Curcumin improved gap junction intercellular communication (GJIC) in B16 cells

GJIC is a mode of cell–cell signaling that enables intercellular communication between adjacent cells through channels of connexins (*Aasen et al., 2016*), and plays a critical role in mediating the effect of drugs used in melanoma therapy (*Aasen et al., 2019*). To study whether curcumin affects the GJIC function of B16 cells, a double-fluorescence dye transfer assay was carried out. In this study, the red fluorescent dye CMTMR and the green dye Calcein were employed. Once entering into the cells, CMTMR remained in the living
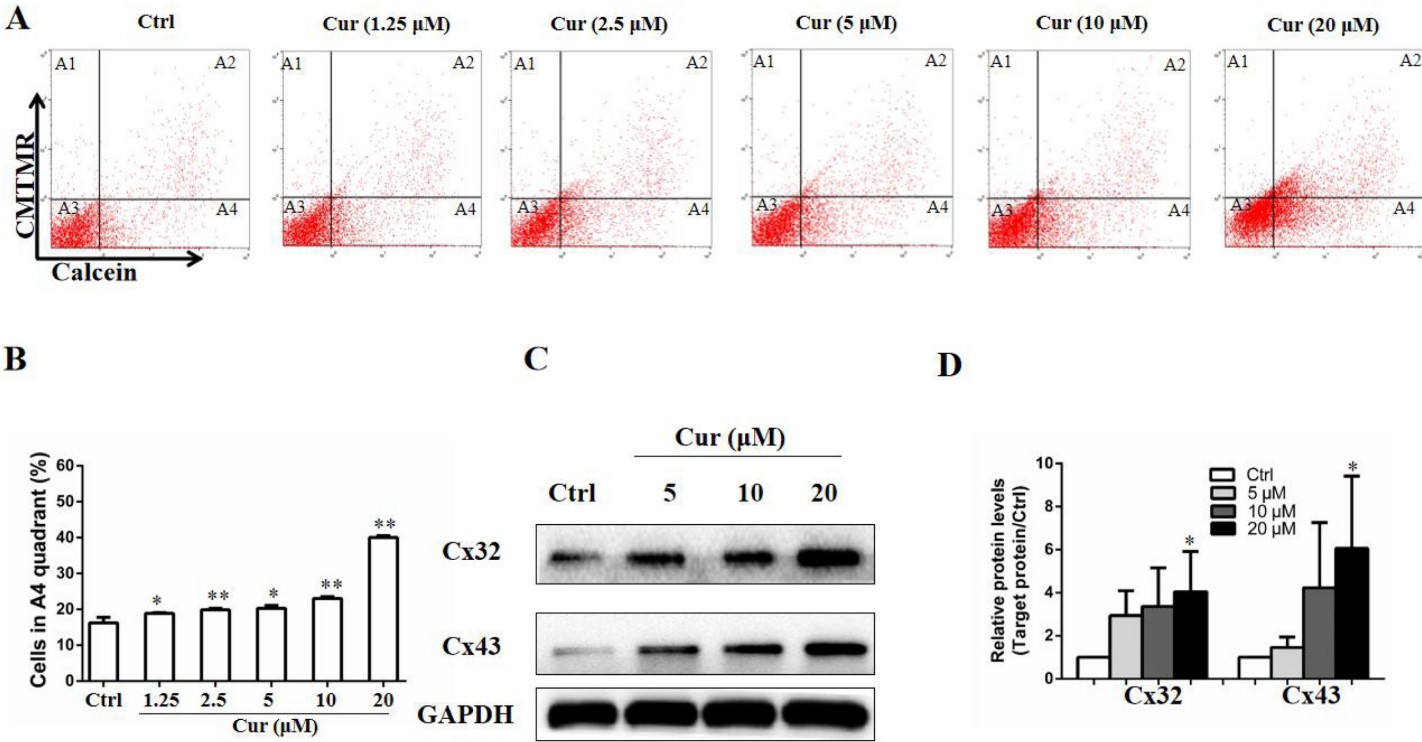

**Figure 2  Curcumin improved GJIC function in B16 cells.** Double-fluorescence dye transfer assay was used to evaluate GJIC function in B16 cells. (A) Cells were sorted by flow cytometry. A1, calcein⁻ and CMTMR⁺ cells, A2, calcein⁺ and CMTMR⁺ donor cells, A3, calcein⁻ and CMTMR⁻ recipient cells, A4, CMTMR⁻ and calcein⁺ cells. (B) The proportion of CMTMR⁻ and calcein⁺ cells in A4 quadrant was quantified. In the parallel study, B16 cells were subjected to curcumin treatment with indicated concentrations for 24 h, followed by harvested for western blot analysis. Protein levels of Cx32 and Cx43 were detected (C) and quantified (D). Data were presented as means ± SD. *$P < 0.05$, **$P < 0.01$ vs. ctrl group.

cells and could not be transferred to adjacent cells, whereas Calcein could be transferred between cells through GJ channel. In this sense, after double-dyed B16 cells were mixed with fluorescence-negative B16 cells by 1:99 and incubated for 4 h, the ratio of cells with green fluorescence to those without fluorescence indicated the function of GJIC. As shown in Figs. 2A and 2B, low dose of curcumin was sufficient to increase calcein transfer. Of note, all concentration tested significantly enhanced the function of GJIC, as indicated by the elevated calcein-positive cells compared to the control group. In addition, the influence of curcumin on connexins, which are essential to GJIC function, was also detected by Western Blot. We found that proteins of Cx32 and Cx43 were both upregulated by curcumin at the concentration of 20 µM (Figs. 2C and 2D).

## Curcumin synergistically improved the inhibitory effect of HSV-TV/GCV in B16 cells

In order to study the effect of HSV-TK/GCV in B16 cells, we constructed B16^TK cells that stably expressed TK gene, and co-cultured these cells with B16^WT cells. As shown in Fig. 3A, the anti-proliferation effect of HSV-TK/GCV in B16 cells was dependent on the ratio of B16^TK cells to B16^WT cells. Since 20% B16^TK cells were sufficient to mediate

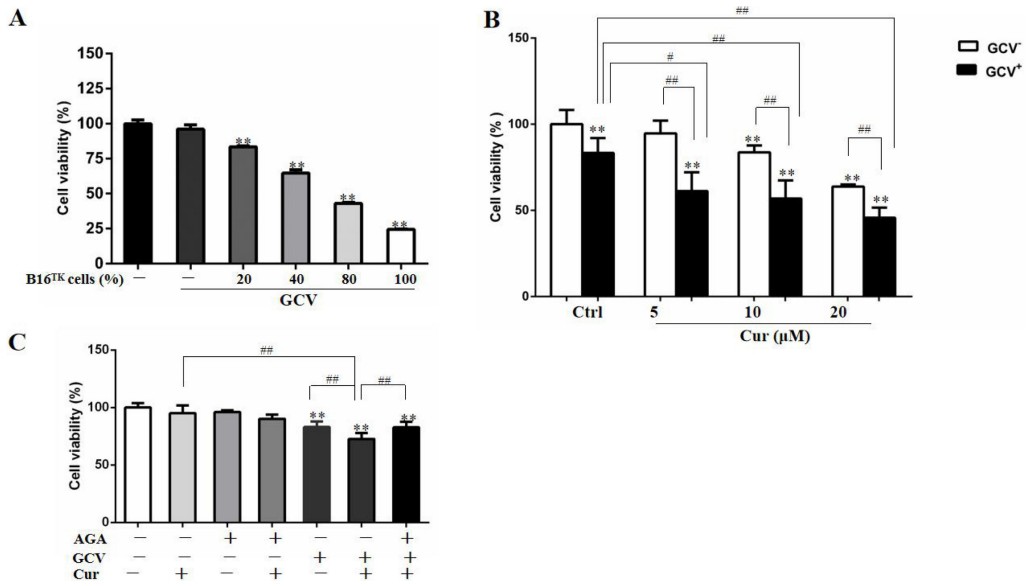

**Figure 3** **Combination of curcumin and HSV-TK/GCV treatment had a synergistic inhibitory effect on the growth of B16 cells.** (A) B16$^{WT}$ cells were co-cultured with indicated proportions of B16$^{TK}$ cells in 96-well plates, followed by treatment with or without 15 μM GCV for 48 h. The cell viability was determined by MTT assay. **$P < 0.01$ vs. GCV$^-$ ctrl group. (B) B16$^{TK}$ and B16$^{WT}$ cells were co-cultured in a ratio of 1:4 in 96-well plates. 24 h later, cells were pretreated with indicated concentrations of curcumin for 1 h, followed by treatment with 15 μM GCV for additional 48 h. The cell viability was detected using MTT assay. **$P < 0.01$ vs. GCV$^-$ ctrl group; #$P < 0.05$, ##$P < 0.01$ vs. the indicated groups. (C) AGA treatment (15 μM) impaired the inhibitory effect of GCV combined with 5 μM curcumin on the mixed cells. **$P < 0.01$ vs. ctrl group; ##$P < 0.01$ vs. the indicated groups. Data were presented as means ± SD.

the inhibitory effect of GCV, we co-cultured the B16$^{TK}$ cells and B16$^{WT}$ cells in 1:4 for the subsequent *in-vitro* studies. By using the MTT assay, we found that 5, 10 and 20 μM curcumin significantly improved the inhibitory effect of HSV-TK/GCV on B16 cell viability (Fig. 3B). The combinational effect was scored by Q value, which was 1.85, 1.43 and 1.16, respectively, indicating the synergistic effect of curcumin and HSV-TK/GCV treatment. Since curcumin inhibited GJIC function, we speculated that the synergistic effect might be mediated by gap junction. Hence, we examined whether the effect was inhibited by AGA, a long-term inhibitor of GJ. As demonstrated in Fig. 3C, AGA pretreatment evidently reversed the inhibitory effect of GCV combined with 5 μM curcumin. The Q value was reduced from 1.34 to 0.68, indicating an antagonistic effect (*Jin, 2004*).

## The bystander effect of HSV-TK/GCV was enhanced by curcumin

The effect of suicide gene therapy is critically associated with its bystander effect (*Duarte et al., 2012*; *Van Dillen et al., 2002*). To examine whether curcumin affected the bystander effect of HSV-TK/GCV treatment, we co-cultured the B16$^{TK}$ cells and B16$^{WT}$ cells at the ratio of 1:4. The B16$^{TK}$ cells and B16$^{WT}$ cells expressed GFP and RFP, respectively. Since GFP is fused with TK and that TK is a nucleoprotein, the B16$^{TK}$ cells expressed GFP in the nuclear. By contrast, RFP in the B16$^{WT}$ cells was mainly expressed in the cytoplasm. The fluorescence images showed that co-treatment of curcumin with GCV markedly

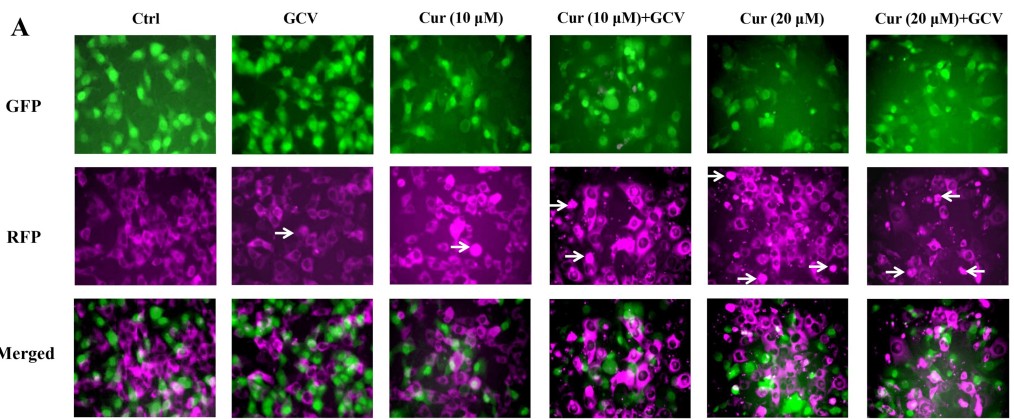

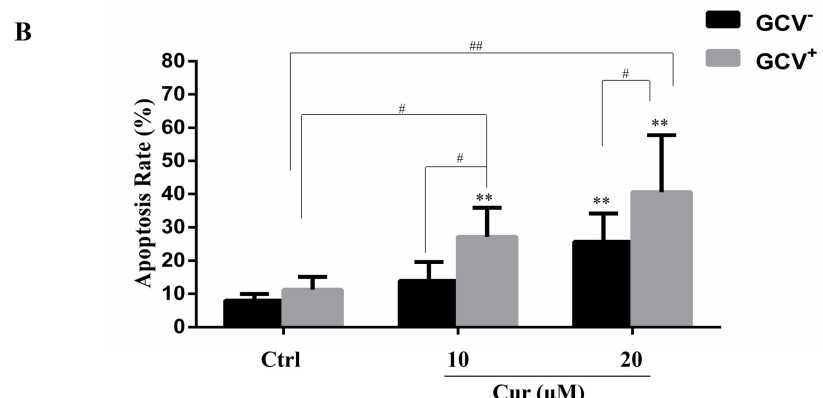

**Figure 4** **The bystander effect of HSV-TK/GCV was enhanced by curcumin.** B16$^{TK}$ and B16$^{WT}$ cells were mixed in 1:4 and cultured in a 6-well plate. Cells were pretreated with or without indicated concentration of curcumin for 1 h, and then exposed to 15 $\mu$M GCV for 72 h. Cells were harvested and stained with Annexin V for detecting the apoptosis rate. (A) GFP fused with TK B16$^{TK}$ cells and RFP in B16$^{WT}$ cells were tracked and images were taken by fluorescence microscopy. The red color was converted into magenta (purple) for better accessibility. White arrows indicate the apoptotic cells. (B) The apoptosis rate of B16$^{WT}$ cells. Data were presented as means $\pm$ SD. $**P < 0.01$ vs. GCV$^-$ ctrl group; $^{\#}P < 0.05$, $^{\#\#}P < 0.01$ vs. the indicated groups.

induced B16$^{WT}$ cells apoptosis, as indicated by pyknosis and cell shrinking. Subsequently, the apoptosis rate of B16$^{WT}$ cells was detected by flow cytometry analysis. The result (Fig. 4B) showed that, compared with the negative control, 20 $\mu$M curcumin alone was able to induce B16 cell apoptosis, as indicated by the B16$^{WT}$ cells stained with Annexin V. Interestingly, compared to the GCV group, the apoptosis rate was significantly elevated by combinational treatment with either 10 or 20 $\mu$M curcumin, suggesting the potentiation of the bystander effect by curcumin. The $Q$ value was 2.71 and 1.69, respectively, which also suggested a synergistic effect.

## Combination of curcumin with HSV-TK/GCV therapy inhibited xenografted melanoma growth

To confirm the synergistic effect of curcumin on HSV-TK/GCV, we performed *in vivo* study using the xenografted melanoma model (Fig. 5A). In order to determine the optimal ratio of B16$^{TK}$ to B16$^{WT}$ cells, we conducted preliminary experiment and injected the cell mixture with 20%, 40%, and 80% B16$^{TK}$ cells, respectively, into the right flank of mice. 7 days later, GCV was administrated for another 7-day period. Then the xenografted tumors were isolated. The result showed that B16$^{TK}$ cells at the percentage of 40% or 80% significantly reduced the tumor weight (Fig. 5B). To better evaluate the combinational effect of curcumin, we mixed B16$^{TK}$ cells with B16$^{WT}$ cells in a ratio of 3:7 in the following *in-vivo* study. The result demonstrated that administration of neither GCV nor curcumin alone affected the size of the xenografted tumor (Fig. 5C). By contrast, the size on day 12 and 13 was significantly reduced by co-administration of curcumin with GCV. On the 14th day, the mice were sacrificed, and the tumor was isolated for measuring volume and mass. As shown in Fig. 5D, while curcumin mildly influenced the weight of xenografted melanoma, GCV exerted an evident inhibitory effect. Notably, combination of curcumin with GCV significantly inhibited the tumor growth, resulting in smaller size (Fig. 5C) and less weight (Fig. 5D). The Q value for the size (1.17) and weight (1.34) suggested that curcumin exerted a synergistic inhibitory effect with HSV-TK/GCV.

## DISCUSSION

Melanoma has been a global concern due to its high mortality and resistant to the conventional therapy such as chemotherapy and radiotherapy (*Bray et al., 2018*; *Ferlay et al., 2019*; *Heo et al., 2016*). Much effort has been made to improve the therapy strategy (*Prado, Svoboda & Rigel, 2019*; *Wrobel, Przybylo & Stepien, 2019*). As a promising therapeutic method, gene therapy has been widely introduced to treat melanoma and other tumors (*Kumar et al., 2016*; *Sotomayor et al., 2002*). However, due to the problem of low efficiency, which caused the insufficient lethality, it is limited in clinical application. Accordingly, using new delivery vectors or taking advantage of the combinational therapy has been a strategy to improve the therapy efficacy (*Bressy, Hastie & Grdzelishvili, 2017*; *Luo et al., 2010*; *Vago et al., 2016*).

HSV-TK/GCV is a well-known suicide gene therapy. In this system, viral thymidine kinase is expressed and consequently metabolizes GCV to mono-phosphorylated GCV. The product will be further phosphorylated by kinases into tri-phosphorylated GCV, an analogue of deoxyguanosine triphostphate, and thus inhibited DNA synthesis, resulting in the tumor cell death (*Duarte et al., 2012*). Of interest, in addition to the suicide effect of this system, the bystander effect was found to be particularly important to the antitumor activity (*Van Dillen et al., 2002*). In this study, we constructed the B16$^{TK}$ cells, which stably expressed the TK gene. To evaluate the effect of HSV-TK/GCV in B16 cells, we mixed B16$^{TK}$ cells and B16$^{WT}$ cells in different ratios. The result showed that B16$^{TK}$ cells dose-dependently induced growth inhibition and bystander effect. Notably, 20% B16$^{TK}$ cells significantly inhibited the cell viability (Fig. 3A), whereas the bystander effect was weak,

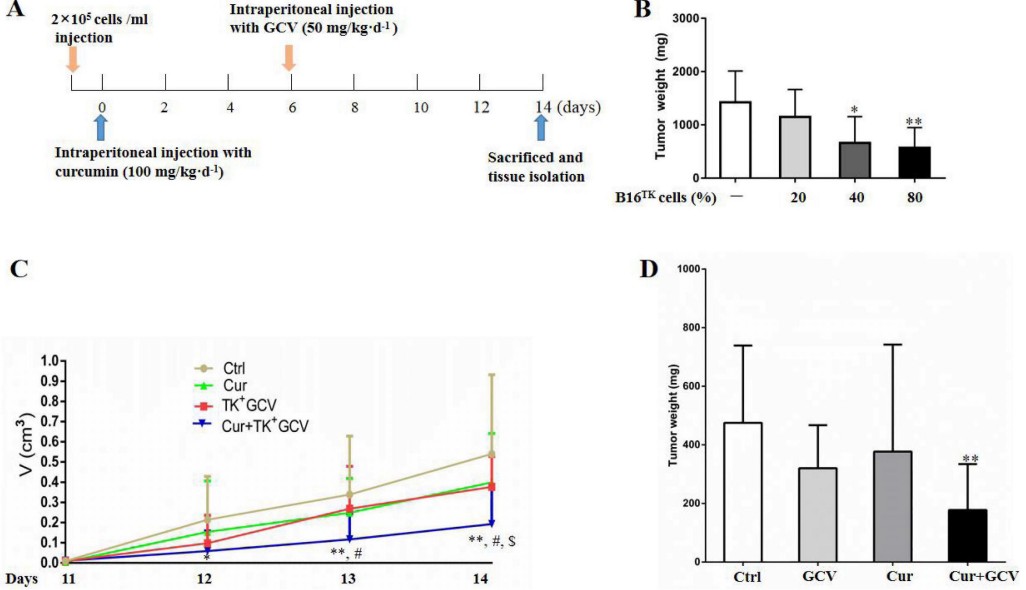

**Figure 5** **Combination of curcumin with HSV-TK/GCV therapy enhanced the inhibitory effect on xenografted melanoma.** (A) Timeline of the *in-vivo* study. (B) B16[TK] and B16[WT] cell mixture with 20%, 40%, and 80% of B16[TK] cells was injected into the right flanks of each C57BL/6J mouse, followed by GCV administration. Then the xenografted tumors were isolated and weighed for analysis. $*P < 0.05$, $**P < 0.01$ vs. the ctrl group. (C–D) Subcutaneous tumors were induced in C57BL/6J mice using B16[TK] and B16[WT] cells mixed at a ratio of 3:7. The mice were divided into 4 groups randomly, and subjected to the indicated treatments. (C) Tumor volume was measured daily since the 11th day till the end. (D) The tumor weight of xenografted tumor. $*P < 0.05$, $**P < 0.01$ vs. the ctrl group; $\#P < 0.05$, Cur + GCV group vs. GCV group; $P < 0.05$, Cur + GCV group vs. Cur group. Data were presented as means ± SD.

as indicated by the less cell death than the supposed 20% B16[TK] cells. Besides, increasing proportion of B16[TK] cells in mixed cells injected into mice resulted in higher impact on tumor growth, as indicated by the xenograft weight (Fig. 5A). These results suggested that both inhibition effect and the bystander effect were influenced by the transfection efficiency. Since the efficiency varies from vector and is always relatively low in humans, we mixed B16[TK] cells with B16[WT] cells in 1:4 in the *in vitro* study and 3:7 in the *in-vivo* study, respectively, to study the synergistic effect of curcumin.

Since the bystander effect is critical to the killing action of HSV-TK/GCV therapy, it is important to reveal the mechanism of the effect and more importantly, to seek ways to enhance the effect. To date, several mechanisms have been demonstrated to mediate the bystander effect of HSV-TK/GCV system, including the involvement of gap junction, E-cadherin regulation and soluble factors release (*Asklund et al., 2003*; *Drake et al., 2000*; *Elshami et al., 1996*; *Garcia-Rodriguez et al., 2011*). Our previous study showed that increased expression of Cx43, the most predominant protein of gap junction, was critically involved in the therapeutic efficacy of melanoma (*Kou et al., 2017*; *Xiao et al., 2018*). In this study, we found that curcumin was able to induce connexins expression and elevate the GJIC function in B16 cells, indicating its potential activity to strengthen the bystander effect.

Curcumin is an active component of turmeric and is recognized as an antitoxic product in response to adverse stimuli such as UV radiation, mechanical injuries and fungi or virus infections (*Ammon & Wahl, 1991*). Recent decades have witnessed the multiple biological activities of curcumin in various diseases (*Aggarwal & Harikumar, 2009*), although some scientists deemed curcumin as a PAINS (pan-assay interference compounds) and an IMPS (invalid metabolic panaceas) because of its potential PAINS-type behaviors such as membrane disruption, aggregation and fluorescence interference (*Nelson et al., 2017*). Actually, the fact that curcumin was limited in clinical use might be mainly because of its poor solubility, low availability and instability (*Aggarwal & Harikumar, 2009*). However, much effect has been made to overcome this problem, and some stable and bioavailable curcumin formulations have been available on market, with scientifically proven benefits (*Antony et al., 2008*; *Bahadori & Demiray, 2017*; *Belcaro et al., 2010*; *Morimoto et al., 2013*).To date, accumulating studies showed that curcumin was effective in inhibiting melanoma cell proliferation and melanoma growth (*Mirzaei et al., 2016*; *Nabavi et al., 2018*; *Zhang et al., 2015*; *Zhao et al., 2016*). Consistently, we found that curcumin inhibited B16 cell viability in both time- and dose-dependent manner. Moreover, curcumin significantly enhanced the inhibitory effect of HSV-TK/GCV in B16 cell viability, and it was a synergistic effect as evaluated by Jin's method (*Jin, 2004*). Of interest, the synergistic effect was abolished by the gap junction inhibitor AGA. However, since curcumin could interact with phospholipids (*Shehzad et al., 2014*) and membrane proteins (*Ingolfsson et al., 2014*) and that gap junction channels are localized at plasma membrane, it is possible that the strengthened effect of curcumin on GJIC might be mediated by some other potential effects of curcumin on membranes. Further study is still required to determine the specific mechanism by loss- and gain-of-function strategy.

Curcumin has been demonstrated to sensitize various types of cancer cells (*Gopinath & Ghosh, 2008*), in which it activates a set of apoptotic pathways, and thus exhibits the anti-proliferation effect (*Dorai et al., 2001*; *Huang et al., 1994*; *Radhakrishna Pillai et al., 2004*). Our previous study showed that HSV-TK/GCV treatment induced cell apoptosis and inhibited tumor growth (*Xiao et al., 2018*). By loading curcumin into the P7L10 peptide micelles to be a more efficient carrier for HSV-TK gene delivery, *Park, Han & Lee (2015)* reported that HSV-tk/R7L10-Cur complex induced C6 rat glioblastoma cell death and reduced the tumor size of xenografted glioblastoma. However, the mechanism was not studied and the dose of curcumin was not demonstrated. In this study, we found that B16 cells co-treated with curcumin and GCV exhibited significantly higher apoptosis rate, and the bystander effect of HSV-TK/GCV was also enhanced by curcumin, as indicated by increased apoptotic B16$^{WT}$ cells. The combinational effect was further confirmed in the *in-vivo* study. Consistently, curcumin synergistically strengthened the anti-cancer effect of HSV-TK/GCV, resulting in reduced xenografted melanoma size and mass. Of note, the inhibitory effect of curcumin or GCV alone on the xenografted melanoma was not statistically significant in this study, which might be attributed to the low proportion of B16$^{TK}$ cells and low bioavailability or instability of curcumin, respectively.

## CONCLUSIONS

In summary, the chemotherapeutic effect of HSV-TK/GCV treatment is delineated using vector-based expression of TK in cancer cells, where it converts the prodrug GCV to lethal GCV triphosphate (*Duarte et al., 2012*). The bystander effect that neighboring cells, without TK expression, also happen to die is essential to its anti-cancer effects (*Van Dillen et al., 2002*). Curcumin is a natural compound that possesses diverse activities (*Prasad et al., 2014*). In the present study, we constructed $B16^{TK}$ cells stably expressed TK gene, and mixed with $B16^{WT}$ cells to study the bystander effects of HSV-TK/GCV, as well as the combinational role of curcumin. Our data disclosed that curcumin synergistically enhanced the anti-proliferation and bystander effects of HSV-TK/GCV. Combinational treatment of curcumin and HSV-TK/GCV effectively inhibited xenografted melanoma growth. These findings provide new insight into the combination therapy of curcumin and HSV-TK/GCV, and may provide a novel therapeutic strategy to increase gene therapy efficacy.

### Funding

This work was supported by grants from the National Natural Science Foundation of China (Nos. 81774028 and 81873146), the Natural Science Foundation of Guangdong Province (2017A030310542), and the Traditional Chinese Medicine Bureau of Guangdong Province (20161075). There was no additional external funding received for this study. The funders had no role in study design, data collection and analysis, decision to publish, or preparation of the manuscript.

### Grant Disclosures

The following grant information was disclosed by the authors:
National Natural Science Foundation of China: 81774028, 81873146.
Natural Science Foundation of Guangdong Province: 2017A030310542.
Traditional Chinese Medicine Bureau of Guangdong Province: 20161075.

### Competing Interests

The authors declare there are no competing interests.

### Author Contributions

- Hong Li conceived and designed the experiments, performed the experiments, analyzed the data, contributed reagents/materials/analysis tools, prepared figures and/or tables, authored or reviewed drafts of the paper.
- Haiyan Du conceived and designed the experiments, performed the experiments.
- Guangxian Zhang performed the experiments, analyzed the data, prepared figures and/or tables.
- Yingya Wu performed the experiments, prepared figures and/or tables.
- Pengxiang Qiu, Jing Guo and Xijuan Liu performed the experiments.

- Jingjing Liu and Lingling Sun analyzed the data.
- Biaoyan Du and Yuhui Tan conceived and designed the experiments, contributed reagents/materials/analysis tools, authored or reviewed drafts of the paper, approved the final draft.

## Animal Ethics

The following information was supplied relating to ethical approvals (i.e., approving body and any reference numbers):

Institute Research Medical Ethics Committee of Guangzhou University of Chinese Medicine provided full approval for this research (20180408).

## Data Availability

The raw data is available in the Supplemental Files.

## Supplemental Information

Supplemental information for this article can be found online at http://dx.doi.org/10.7717/peerj.7760#supplemental-information.

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
