# Peer review of "Curcumin plays a synergistic role in combination with HSV-TK/GCV in inhibiting growth of murine B16 melanoma cells and melanoma xenografts"

_PeerJ, doi:10.7717/peerj.7760_

## Round 0.1 · original submission · Major Revisions

Please address the critiques, paying serious attention to the comments of reviewer #1.

Reviewer 1 ·

Basic reporting

See the general comments

Experimental design

See the general comments

Validity of the findings

See the general comments

Additional comments

In this manuscript, Li and co-workers applied curcumin to HSV-TK/GCV gene therapy and found that curcumin can enhance the bystander effect mediated by gap junction. Furthermore, the in vivo study showed promising inhibition of the xenograft growth. Based on the results from in-vitro and in-vivo studies, the authors claimed that the findings provided new insight into the combination therapy and might provide a novel therapeutic strategy to increase gene therapy efficacy. The paper is well written, and the experimental results are solid and convincible. However, the manuscript is lacking discussion on the drawbacks of curcumin which has been demonstrated as both a PAINS (pan-assay interference compounds) and an IMPS (invalid metabolic panaceas).

Curcumin is a PAINS because it shows positive results in most drug discovery assays but fails to advance as drug leads. So far, curcumin has not been successful in any clinical trial. Moreover, the current research has found that curcumin exhibits some confounding behaviors including covalent labeling of proteins, metal chelation, redox reactivity, aggregation, membrane disruption, spectral interference, and structural decomposition (J. Med. Chem. 2017, 60, 1620−1637). There are a few questions the authors need to address in the manuscript.

1. The solubility of curcumin in the water-based buffers is very poor and is pH- and temperature-dependent. Typically, the highest concentration of curcumin in water is ~10 μM. In practice, curcumin can only dissolve in water at acidic pH and high temperature. Although there are some studies reported the dissolving of curcumin in water at ambient temperature with higher concentration, it turned out that the curcumin either degraded or aggregated during the dissolving process. The authors need to show the details of the preparation of curcumin solutions and provide the evidence that the curcumin is the actual major components in the solutions.

2. Curcumin is very unstable and the T1/2 is less than 5 min both in vitro and in vivo (J. Pharm. Biomed. Anal., 1997, 15, 1867−1876 and J. Chromatogr. B: Anal. Technol. Biomed. Life Sci., 2007, 853, 183−189). Therefore, the positive assay results may due to other products from degradation rather than curcumin.

3. Curcumin forms chemical aggregates (colloidal) under common biochemical assay conditions. The previous study showed that a critical aggregation concentration is ~17 μM by DLS (ACS Chem. Biol. 2015, 10, 978−988). The enzymes involved in the biochemical assays may interact with the chemical colloids, thereby changing the actual protein concentrations which leads to false positive assay results. The authors should check the aggregation level of the curcumin sample before the assays (i.e. by DLS). Otherwise, the authors should discuss the reasons if they think the chemical aggregation won’t affect the assay results.

4. The authors claimed that the bystander effect was enhanced because of the improvement of GJIC by curcumin. However, several studies have demonstrated that the curcumin has strong interaction with phospholipids and alter bilayer properties (ACS Chem. Biol., 2014, 9, 1788−1798). Also, curcumin can interact with membrane proteins and change their functions. The authors need to discuss whether the double-fluorescence dye transfer assay in this study can rule out the possibility of membrane malfunction caused by curcumin.

5. In another study (Pharm. Res., 2015, 528–537), the researchers have already tested the effect of curcumin on HSV-TK/GCV gene therapy as well. They designed a deliver peptide to transport both the HSV-TK/GCV gene and curcumin simultaneously to the target spots to overcome the poor bioavailability, solubility, and stability of curcumin. They also found that curcumin induced cell death efficiently both in vitro and in vivo which is the same as the conclusion drawn in this manuscript. Please cite this paper and discuss what is novel in your study compared with that one.

6. The reference didn’t organize very well. There is a total of 38 papers cited in the manuscript, but there are only 37 papers in the reference section. Please correct this.

Reviewer 2 ·

Basic reporting

Basic reporting is clear and unambiguous.
Sufficient literature is cited.
The manuscript has a professional structure, figures, and tables.
The manuscript has relevant results.

Experimental design

The original primary research is within the aims and scope of the journal.
The research question is well defined and relevant.
The methods are described in sufficient detail.

Validity of the findings

The results are novel.
The underlying data have been convincingly provided with robust, statistical controls.
The conclusions are well stated.

Additional comments

Line31: Please change (CX)32 to CX32
Line 73: Please use the standard scientific language for Curcuma longa (italicize it).
Line 200: Please show the data for antiproliferation effect by mixing the B16TK and B16WT cells.

---

## Round 0.2 · accepted · Accept

Since all the critiques were adequately addressed and since the manuscript was amended accordingly, I am pleased to accept your revised article.

Reviewer 1 ·

Basic reporting

See the general comments

Experimental design

See the general comments

Validity of the findings

See the general comments

Additional comments

In the resubmission, the authors discussed the drawbacks of curcumin and addressed the questions from previous reviewers properly. This study demonstrated that the curcumin was able to enhance the bystander effect of HSV-TK/GCV system by improving the gap junction intercellular communication function. Although the degradation of curcumin was inevitable and quantitative control of curcumin concentration in the in-vivo study was infeasible, the authors overcame these issues by careful design of the control experiments. Overall, the resubmission is a well-written and solid paper, I recommend that the manuscript should be accepted for publication in its present form.